# Insights on Proteomics-Driven Body Fluid-Based Biomarkers of Cervical Cancer

**DOI:** 10.3390/proteomes10020013

**Published:** 2022-04-29

**Authors:** Amrita Mukherjee, Chinmayi Bhagwan Pednekar, Siddhant Sujit Kolke, Megha Kattimani, Subhiksha Duraisamy, Ananya Raghu Burli, Sudeep Gupta, Sanjeeva Srivastava

**Affiliations:** 1Department of Biosciences and Bioengineering, Indian Institute of Technology Bombay, Mumbai 400076, India; 184303011@iitb.ac.in; 2Department of Life Science, Ramnarain Ruia Autonomous College, Mumbai 400019, India; chinmayipednekar01@gmail.com; 3Department of Chemistry, Indian Institute of Technology Bombay, Mumbai 400076, India; 19b030028@iitb.ac.in; 4Undergraduate Department, Indian Institute of Science, Bengaluru 560012, India; meghak@iisc.ac.in; 5Department of Human Genetics and Molecular Biology, Bharathiar University, Coimbatore 641046, India; subhikshaduraisamy@gmail.com; 6Department of Chemical Engineering, Indian Institute of Technology Bombay, Mumbai 400076, India; ananya.burli@iitb.ac.in; 7Advanced Centre for Treatment, Research and Education in Cancer, Tata Memorial Hospital, Mumbai 400012, India; sudeepgupta04@yahoo.com

**Keywords:** LC-MS/MS—liquid chromatography coupled with tandem mass spectrometry, DIGE—difference gel electrophoresis, CVF—cervicovaginal fluid, MALDI—TOF—matrix-assisted laser desorption/ionization—time of flight, CSCC—cervical squamous cell carcinoma, iTRAQ—isobaric tags for relative and absolute quantification, ELISA—enzyme-linked immunosorbent assay, HPV—Human Papillomavirus, CIN—cervical intraepithelial neoplasia, CaCx—cervical cancer, MRM—multiple reaction monitoring, PRM—parallel reaction monitoring, LFQ—label-free quantification

## Abstract

Cervical cancer is one of the top malignancies in women around the globe, which still holds its place despite being preventable at early stages. Gynecological conditions, even maladies like cervical cancer, still experience scrutiny from society owing to prevalent taboo and invasive screening methods, especially in developing economies. Additionally, current diagnoses lack specificity and sensitivity, which prolong diagnosis until it is too late. Advances in omics-based technologies aid in discovering differential multi-omics profiles between healthy individuals and cancer patients, which could be utilized for the discovery of body fluid-based biomarkers. Body fluids are a promising potential alternative for early disease detection and counteracting the problems of invasiveness while also serving as a pool of potential biomarkers. In this review, we will provide details of the body fluids-based biomarkers that have been reported in cervical cancer. Here, we have presented our perspective on proteomics for global biomarker discovery by addressing several pertinent problems, including the challenges that are confronted in cervical cancer. Further, we also used bioinformatic methods to undertake a meta-analysis of significantly up-regulated biomolecular profiles in CVF from cervical cancer patients. Our analysis deciphered alterations in the biological pathways in CVF such as immune response, glycolytic processes, regulation of cell death, regulation of structural size, protein polymerization disease, and other pathways that can cumulatively contribute to cervical cancer malignancy. We believe, more extensive research on such biomarkers, will speed up the road to early identification and prevention of cervical cancer in the near future.

## 1. Introduction

More than 60% of an adult human body is made up of fluids, with extracellular fluids (plasma, serum, mucus, saliva, urine, bile, etc.) accounting for one-third of total body water content [1]. Body fluids are becoming a more appealing subject for clinical study or diagnosis in the new era, thanks to advancements in omics technologies. This has occurred for a variety of reasons, including the following: such fluids are constantly generated and are easily accessible making multiple sampling possible, and convenient acquisitions of the same without the need for invasive procedures such as a biopsy [2]. As a result, patients have a less unpleasant and minimally invasive experience. Furthermore, biological fluids have the potential for building low-cost prognostic/diagnostic test kits [3]. Extensive proteomic studies of such body fluids yield biomarker information with exceptional diagnostic value. Quantitative proteomics, on the other hand, offers information on the proteins that are present or absent in certain clinical sample groups, as well as assists us in identifying differentially expressed proteins in illness vs. healthy circumstances [3].

Quantitative label-free proteomics involving liquid chromatography coupled with tandem mass spectrometry (LFQ-LC-MS/MS) is one of the widely used approaches for such proteome-based biomarker identification. This approach is capable of profiling a huge number of samples and analyzing them without the need for labelling [4]. These characteristics make LC-MS/MS an excellent choice for biomarker discovery. DIGE is another technique where several samples can be run on the same gel, each one identified by a fluorescent dye [5]. The non-invasive nature of body-fluid testing and subsequent clinical proteomics analysis, as previously stated, opens the path for biomarker-based early cancer diagnosis. Proteomic biomarkers are majorly categorized into four types: *Diagnostic biomarkers*, allow the early detection of cancer; *Prognostic biomarkers*, offer information about the disease’s expected progress; *Therapeutic biomarkers*, are proteins that can be exploited as a therapeutic target using drugs/small molecules; lastly, the *predictive biomarkers* basically predict a patient’s reaction to targeted therapy and thereby establish subpopulations of patients who are likely to benefit from that. In this review, we will mostly discuss diagnostic biomarkers from the literature available in the field of cervical cancer.

Proteomics-based technologies are being used to uncover biomarkers in a variety of diseases, and they are promising because the amount of protein, its structure, and its functions can all be utilized as indicators to improve early disease identification or prediction [6]. When it comes to body fluids, the abundance of biomarkers will be highest in those originating from the proximal tumour microenvironment [5]. Blood is the most prevalent body fluid used for biomarker identification, followed by saliva, urine, CSF, and then microenvironment-specific body fluids [5]. Cervical cancer patients are divided into four major stages (I to IV) by FIGO (The International Federation of Gynecology and Obstetrics), where each stage is again divided into multiple sub-stages. In stage I, the tumour is very small, and only confined to the cervix. In stage II, it starts spreading beyond the uterus, present in the upper two thirds of the vagina. In stage III and beyond, the tumour becomes aggressive and spreads to nearby pelvic lymph nodes and distant organs. Cervical carcinoma beyond stage IB2 is called locally advanced cervical cancer or LACC [7]. When it comes to cervical cancer, current routes of therapy for FIGO stages IIA and above involve radiotherapy along with administration of Cisplatin [8]. The hunt for novel therapeutics becomes especially relevant in this case, which manifests therapy resistance (radioresistance and chemoresistance) in its later stages. Furthermore, proteomic approaches can be extremely helpful for understanding the modes of therapy resistance by comparing the protein profiles of drug-resistant cells and drug-sensitive cancer cells [9]. Given the plethora of information that can be derived from proteomics methodologies based on body fluids, this route has a lot of promise for creating new therapies for cervical cancer. In this review, we will present various types of body fluids that are useful in cervical cancer diagnosis and will discuss the possible protein biomarkers reported from them (blood, serum, cervical-vaginal fluid or CVF, amniotic fluid, urine, etc.). Next, we will discuss the types of body fluids that have been used to find CaCx biomarkers in the past, as well as the obstacles of sample collection, along with the benefits and drawbacks of each. Finally, we will present a metadata analysis of three different studies based on cervicovaginal fluid (CVF) proteomics to provide a glimpse into cumulative biomarkers and various cellular pathways found to be important for cervical cancer. This will help elucidate the scope of CVF, a body fluid proximal to the cervicovaginal environment, to be taken as a prominent diagnostic fluid in the future.

### 1.1. Proteomics-Driven Biomarker Discoveries from Diverse Body Fluids

Clinically relevant biomarker research, which is fueled by omics-based technologies, particularly shotgun proteomics, has already made significant advances. Such omics-based (genomics and proteomics) techniques hold the promise of an unbiased discovery route and are considerably more systematic than other approaches, which is one of the reasons why these omics technologies are gaining so much popularity [10]. Proteomics driven clinical biomarker discovery happens in two phases: The discovery phase, where the shotgun approach is used to test and characterize protein biomarkers in a larger cohort; and the validation phase, where the selected candidates are validated in a blinded manner using targeted proteomics approaches [3] (Figure 1).

There have been plenty of studies focusing on body fluid-based biomarkers for different types of malignancies throughout the years (Table 1). The demand for non-invasive and accessible diagnostics has sparked a lot of interest in them [11]. Moreover, biomarkers in body fluids show the presence of microRNA, RNA, DNA, proteins, lipids, and circulating tumour cells (CTCs); and combining biomarkers from diverse body fluids might reduce false negatives and perhaps offer us a superior understanding of tumour subtypes [11]. The elimination of high abundance proteins such as albumin, which hide the presence of biomarkers that are generally low in number, is one of the initial stages of body fluid proteomic analysis [1]. Although plasma and serum are valuable resources for biomarker identification, the success rate is poor due to limited repeatability, considerable variability from sample to sample, and a high dynamic range of serum proteins [3,12]. More localized body fluids such as the cerebrospinal fluid or the cervicovaginal fluid can lead to biomarker identification from specific tumour microenvironments in organs such as the brain and the cervix, respectively. Such organ-specific body fluids can help us identify biomarkers directly associated with the respective cancer and provide a more specific and sensitive biomarker [2,13].

**Table 1 proteomes-10-00013-t001:** Types of diagnostic body fluid biomarkers identified from various cancer studies.

Body Fluid Type	Cancer/Disease Type	Cohort Used	Biomarkers Find	Methodology Applied	Reference
Blood	HBV induced Hepatocellular carcinoma (HCC)	22 patients affected by HBV induced HCC and 22 healthy controls	Alpha-fetoprotein (AFP)	Enzyme-linked immunosorbent assay (ELISA) and SPSS for statistical analysis	[14]
Urine	Pancreatic cancer	[I] Healthy: 87 individuals; Pancreatic cancer: 192 individuals [II] Healthy: 87 individuals; Pancreatic cancer: 71 individuals	Lymphatic vessel endothelial hyaluronan receptor-1 (LYVE-1), regenerating gene-1-alpha (REG-1-alpha) and trefoil factor-1 (TFF-1)	GeLc/MS/MS analysis; biomarker validation was conducted via ELISA and a multiple logistic regression model was applied to a training dataset of 488 urine samples in a multicentre cohort	[15]
Serum	HNSCC	Healthy: 10 individuals; HNSCC: 39 individuals (37 men and 2 women)	MMP 13	A two-site sandwich ELISA assay was used to evaluate the markers	[16]
Serum	Endometrial cancer	174 endometrial cancer patients. Samples were taken at four points: (i) primary diagnosis, (ii) post-surgery, (iii) follow-up, and (iv) at recurrence	HE4, CA 125 (cancer antigen 125)	Levels of biomarkers were measured using chemiluminescent enzyme immunoassay (CLEIA)	[17]
Plasma	Laryngeal squamous cell carcinoma (LSCC)	22 patients diagnosed with advanced LSCC and 21 healthy controls	miR-31-3p and miR-196a-5p	RT-qPCR was used to estimate the presence of biomarkers. Tissue and plasma samples were correlated and the two miRNAs were found to be upregulated in both tissue and plasma samples	[18]

Tissue-specific or organ-specific body fluids have altered proteins unique to the tissue/organ which are secreted from the cells present in that microenvironment and hence have a great potential for us to discover extremely specific prognostic and diagnostic biomarkers that can be distinctive for a certain cancer type [5]. In the case of pancreatic cancer, there is a pre-existing diagnostic body fluid-based biomarker, carbohydrate antigen 19-9 (CA 19-9), which is present in serum, is not used for detection but for follow-up in patients who are already diagnosed [19]. One of the main reasons stated as to why it is not used for early detection of pancreatic cancer is because it does not have adequate sensitivity and specificity [20]. Lymphatic vascular endothelial hyaluronan receptor-1 (LYVE-1), regenerative gene-1-alpha (REG-1-alpha), and trefoil factor-1 (TFF-1) are potential biomarkers for the detection of early-stage pancreatic cancer isolated from urine [15,19]. Body fluid-based biomarkers also have great potential for the early detection of head and neck squamous cell carcinomas (HNSCCs). The most promising biomarkers found till now are metalloproteinases (MMPs) from serum [11,16], interleukins 6 and 8 (serum) [21], and cytokeratin 17 (CK-17) from mucus [22,23]. Prognosis and diagnosis via body fluid-based biomarkers are non-invasive and can be a faster method than biopsies. They can normalize testing for more stigmatized cancers such as prostate or cervical cancer and this normalization can help in the early detection of the disease. If a specific biomarker has to be discovered, proteomic techniques can be proved to be precise, even when those protein(s) in clinical samples are dominated by more abundantly present proteins.

### 1.2. Different Types of Diagnostic Body Fluids in Cervical Cancer

For a body fluid to be considered as a source of potential biomarkers of cervical cancer, it needs to either originate from the cancerous tissue or it has to be in close contact with the cancerous tissue (peritumoural tissue), so that it carries over the altered protein composition of the tissue mass in itself. In the latter, the body fluids gain the tissue-specific proteins as the diseased cells are shed into the body, and thus, it carries the altered protein expression profiles of the disease representing the changes in the molecular and cellular networks [5]. The most common and widely used body fluids for biomarker discovery in cervical cancer are plasma and serum from blood and tissue cells. A disadvantage of using plasma and serum is the sheer number of proteins from all different types of tissue microenvironments along with the already abundant proteins present such as albumins, immunoglobulins (Igs), alpha-1-antitrypsin (A1AT), fibrinogens, and haptoglobin (HG) [24]. Thus, these body fluids need to undergo immunodepleting events to reduce the intensity of the common and abundant proteins so that the detection of proteins with lower abundance is enhanced [25]. Another common body fluid for cervical cancer biomarker discovery is cervical mucus. Cervical mucus is a mixture of proteins in both aqueous and glycoprotein phases. The cervical mucus composition changes with hormonal changes during the menstrual cycle, menopause, and due to the use of hormonal contraceptives. Just like plasma and serum samples, immunodepletion should be essentially conducted with the mucus samples as well [25,26]. Urine is also commonly used to identify biomarkers for cervical cancer. In general, the proteomic composition of urine is less complex than that of serum. Urine is also more thermodynamically stable than serum, making it easier to analyze and a promising candidate for biomarker discovery [24]. Cervicovaginal fluid is another promising body fluid that has been investigated for cervical cancer biomarkers previously. Cervicovaginal fluid (CVF) consists of cervical mucus, vaginal wall transudate, exfoliated cells, secretions from the endometrium and the oviducts, and the metabolic by-products of vaginal microflora [27]. Just like the cervical mucus, the composition of the CVF changes with the dynamic hormonal cycles [28]. The biomarkers obtained from this body fluid can be important as they are closely associated with the cervical and vaginal microenvironment [29]. Although the research done is not very extensive, it has great potential for the identification of cervical cancer-specific biomarkers. All the above-mentioned body fluids can be great tools in identifying cervical cancer biomarkers and can even lead us to targets for drugs and other therapies. With a panel of biomarkers from different body fluids, it could be possible to judge the severity of the cervical carcinoma as well as prescribe a personalized therapy for the patient, making their treatment more effective.

## 2. Proteomic Biomarkers Identified from a Variety of Body Fluids of Cervical Cancer Patients

### 2.1. Blood (Plasma) Based Biomarkers

Human blood is composed of numerous vital components, each of which serve a specific purpose. Several components of blood primarily include plasma, human serum, red blood cells, and white blood cells. Blood plasma is the most appealing alternative for biomarker mining since it is the medium of transport and dispensing site for all cellular products in the body. Apart from the most abundant proportion of protein with function in blood, like albumin, the plasma proteome is dynamic in nature, encompassing immunoglobulins, receptor ligands, leakage secretions of tissues, aberrant secretions, etc. [30]. Plasma is the body fluid most conventionally used for diagnosis of various unregulated physiological conditions. Hence discovering plasma-based biomarkers for cancer could aid in early detection (Table 2) [31,32,33,34]. Studies on the discovery of plasma-based biomarkers for cervical cancer appear to be infrequent and less extensive in comparison to other cancer types; such biomarker identification studies frequently use proteomic and metabolomic methods [13,35,36,37,38,39,40]. Although the simplicity of use and handling of histopathological techniques is favorable, the discovery of biomarkers in plasma necessitates the enrichment of low abundance proteins. The major methods employed in plasma proteome-based workflow include separation by 2D-DIGE, MALDI—QTOF MS/MS analysis (by label-free or label-based quantitation or targeted proteomics), biomarker validation by ELISA, and statistical analysis [41,42]. Using a multiplex proximity extension assay, a very recent study has identified a signature of 11 proteins (PTX3, ITGB1BP2, AXIN1, STAMPB, SRC, SIRT2, 4E-BP1, PAPPA, HB-EGF, NEMO, and IL27) that can distinguish between invasive cervical cancer patients and healthy controls. When compared to population controls, there was no variation in the abundance between samples obtained before and after therapy, showing that the protein profile can be proven as one of the most informative for developing diagnostic biomarkers [36]. Guo and colleagues conducted a comparative plasma proteome analysis of samples taken from 22 healthy women and 26 women with early-stage cervical squamous cell carcinoma (CSCC) in China, where cervical cancer is considered to be widespread [13]. A thorough inspection by 2D-DIGE of low abundance enriched proteins followed by MS analysis identified three major proteins correlated with cervical carcinogenesis: ApoA1 (Apolipoprotein A1), ApoE (Apolipoprotein E), and CLU (Clusterin). ApoA1 and ApoE are both lipoprotein components of HDL. ApoA1 is assumed to be inherently involved in apoptosis promotion by the MAK pathway and involved in anti-proliferative and anti-metastatic effects through both innate and adaptive immune responses [43]. Though expression status varies with different cancer types, suppressed expression is highly associated with increased risk of metastasis and could provide good prognostic markers for cervical cancer. ApoE is a multifunctional protein that plays a role in lipid transport and lipoprotein homeostasis, as well as immunological responses, cell proliferation, and smooth muscle cell migration [44]. ApoE is associated with metastasis and tumor growth, progression and staging in various types of cancer [45] and increased ApoE levels directly proportional to the plasma HDL levels indicating an increased risk of breast cancer [46]. Thus, ApoE could be a potential biomarker to indicate the invasiveness level and metastasis state of cervical cancer including cell adhesion, programmed cell death, immunological complement cascade, and lipid transport.

**Table 2 proteomes-10-00013-t002:** Biomarkers identified from different body fluids of cervical cancer patients.

Body Fluid Type	Methodology and Protocol	Cohort	Key Findings	Extra Comments (Merits/Demerits)	References
Plasma	2D-DIGE separation (stained with cytidine dyes); MALDI—TOF/TOF MS analysis; ELISA for biomarker validation and statistical analysis.	Healthy: 22 individuals; early-stage CSCC (cervical squamous cell carcinoma) patients: 22 individuals.	ApoA1, ApoE and CLU were validated by ELISA as prognostic markers. ApoA1 was downregulated and ApoE and CLU were upregulated in CSCC.	Identifying individual or panel of potential biomarkers at a treatable stage.	[13]
Plasma	2D-DIGE (silver staining); MS/MS (MALDI-TOF) to identify DEPs, and further validation by ELISA and statistical analysis by ANOVA.	Healthy: 40 individuals; CSCC and CIN patients: 80 individuals.	Cytokeratin 19 is upregulated in both the CIN 3 and CSCC IV conditions and tetranectin downregulated in CSCC.	Identification of DEPs along different stages of cervical cancer progression helps in understanding and prognosis of cancer.	[38]
Serum	Weak cation method, exchange chromatography fractionation in conjunction with MALDI-TOF spectrometry, liquid chromatography-electrospray ionization tandem mass spectrometry, and enzyme-linked immunosorbent assay (ELISA).	Healthy: 50 individuals; patients before surgery: 39; patients after surgery: 28.	The three peaks (m/z: 2435.63, 2575.3, and 2761.79 Da) may serve as predictive serum biomarkers for cervical cancer (CC).	Each patient group has obvious variation as the combined effect of age, stage, and tumor type reduces the power of marker detection.	[47]
Serum	In-house developed ELISA with linear peptide envelope antigens derived from TAAs.	Healthy: 28 individuals; CIN I: 28 patients; CIN II: 30 patients; CIN III: 31 patients; cancer: 31 patients.	Survivin, TP53, CyclinB-1 and ANXA-1, c-myc proteins were found differentially expressed in various cancer groups which could be potential biomarkers.	NA	[48]
Serum	Immunoaffinity chromatography, SDS-PAGE, and in-gel digestion, LC-MS/MS; pooled serum sample expression was determined by Western blot.	Healthy: 16 individuals; cervical cancer patients: 31 Individuals.	A1AT, PYCR2, TTR, ApoAI, VDBP, and MMRN1 were expressed considerably differently in serum samples from healthy controls and cervical cancer patients.	VDBP is primarily generated and secreted by the liver and is the principal transporter of vitamin D and its metabolites to target organs.	[49]
Serum	iTRAQ, label-free shotgun mass spectrometric quantification, and targeted mass spectrometric quantification.	For serum pooling and iTRAQ labelling: healthy set_1: 10; healthy set_2: 7; cancer early stage: 9; cancer late-stage: 7; For Label-Free NanoChip-LC/MS Quantification and Targeted MRM Analysis: healthy controls- cervical intraepithelial neoplasia- cancer early stage- cancer late-stage-ovarian cancer.	Patients and healthy controls showed significant changes in abundance of alpha-1-acid glycoprotein 1, alpha-1-antitrypsin, serotransferrin, haptoglobin, alpha-2-HS-glycoprotein, and vitamin D-binding protein.	NA	[50]
Mucous	SELDI-TOF (surface-enhanced laser desorption and ionization-time of flight mass spectrometry).	Samples were collected from women attending urban hospital colposcopy clinics who were enrolled as a part of the study of cervical neoplasia. Samples were collected at the time of colposcopy by absorption into two Weck-Cel^®^ sponges from 2–6 women matched for ages and races.	Annexin, tropomyosin, 14-3-3 sigma, calreticulin, and anterior gradient protein were identified.	The short sample size and inaccuracy of sample collecting techniques lowered the number of proteins discovered	[28]
Mucous	Screening by LC-MS (liquid chromatography-mass spectrometry and gene ontology to predict functions. Differentially expressed proteins in the cervical adenocarcinoma patients and the controls. were conducted using the iTRAQ.	Healthy: 3 individuals; endocervical adenocarcinoma: 3 patients; in situ adenocarcinoma: 3 patients.	The top differentially expressed proteins were APOB, FINC, K1C13, SPTA1, CATA, K2C4, PERM, CO4B, A1AT, CFAH, A2ML1. For AIS: EA they were, PP2AA, HBG2, SBP1, APOC3, IGA2, HSP27, PERM, FINC. AIS: Control patients, the differentially expressed proteins were F10A5, SKP1, HBG2, PNCB, KPYM, SPR1A, MRS.	Although there are two different types of cervical cancer samples, the sample size was very small.	[51]
Menstrual fluid	Genomic DNA was extracted from the menstrual blood collected on a napkin using a QIAmp DNA Mini Kit. Two rounds of PCR reaction using My11 and My09 primers for HPV detection. Fischer’s exact test to examine the association between the distribution of genotypes or alleles for the TAP polymorphisms.	Control: 137 individuals; CIN3, CIN1, CIN2: 265 patients.	TAP1 I333V and TAP1 D637G were detected in the menstrual blood samples. The genotypes AA, AG, and GG were detected at each polymorphic site in the patients and the risk of developing high-grade cervical neoplasia was reduced for the AG and GG phenotypes as compared to the AA genotype. The risk of developing high-grade CIN was reduced in the patients that had a G allele than in those with an A allele.	The findings in the study have high specificity, sensitivity, and positive predictive value for the HPV virus and have received positive responses from over 5000 women.	[52]
Cervicovaginal fluid	Label-free quantification method based on LC-MS/MS method followed by ELISA. The PLS-DA model for further statistical analysis.	Development set—healthy: 10 individuals; LSIL: 10 individuals; HSIL: 10 individuals; cancer: 10 individuals Validation set—healthy: 14; LSIL: 8; HSIL: 6; cancer: 5.	ACTN4, VTN, ANXA1, ANXA2, CAP1, MUC5B and PKM2 from the 27 differentially expressed proteins have been indicated as promising biomarkers for cervical cancer.	The comparatively high number of samples gives better and more accurate results and reduces the chances of false biomarker discovery. The samples were also better classified into further four subgroups providing a comparison basis amongst the four groups.	[53]
Cervicovaginal fluid	Label-free quantification method based on LC-MS/MS method followed by ELISA. Significant proteins were determined using normalised spectra abundance factor values (NSAF values). Chi-squared test to determine the exclusivity of the protein and Unpaired Student’s *t*-test to analyse the ELISA results.	Healthy: 6 individuals; precancerous: 6 individuals.	They determined protein biomarkers for the precancerous state of cervical cancer. They found 12 proteins, including ACTN4 and PKM2.	There is a significant statistical analysis conducted to determine the significant proteins among the ones discovered after the ELISA results.	[54]
Urine	Label-free quantification- UPLC-MS/MS analysis of pooled samples protein-protein interaction (STRING), pathway enrichment analysis, and molecular functions from KEGG and GO. Sensitivity as potential biomarkers tested by Western blotting and statistical analysis like logistic regression, ROC and AUC.	Healthy: 13 individuals; cervical cancer: 24 individuals.	Five Proteins with molecular weight >100 kDA were identified as potential biomarkers—LRG1, MMRN1 (upregulated), S100A1, CD44, SERPIN 33 (downregulated).	Rather than conventional gel-based MS analysis, non-gel based LFQ-MS analysis could aid in finding the low molecular weight potential biomarkers present in trace amounts in urine.	[55]
Urine	2-DE and MALDI-MS and MS/MS analysis, validation by nano LC-MS analysis (LTQ Orbitrap XL ETD mass spectrometer), immunoblotting and statistical analysis.	Healthy: 31 individuals; cervical cancer: 42 individuals.	PCDH8, ARNTL2, serum albumin and Endorepellin, C-terminal domain V of perlecan were found to be differentially expressed. Only endorepellin L3 fragment showed significantly elevated expression levels.	Pre-processing of samples prior to gel-based applications could reduce interference in urine.	[56]

NA: No information added in the table.

A similar plasma proteome study carried out by Looi et al. in 2009, aimed to associate differences in plasma proteome at different stages of tumorigenesis, especially aiming to identify unique biomarkers for CIN 3 (cervical intraepithelial neoplasia) and the CSCC stage IV of cervical cancer. The range of samples were collected from all grades and stages of CIN and CSCC, along with healthy individuals’ samples. Proteomic analysis from patients with CIN 3 and CSCC stage IV across all samples, revealed 18 differentially expressed proteins, the majority of which were acute-phase proteins, transport proteins, coagulation factors, cytolysis inhibitor proteins, and structural proteins. Further MS analysis and validation by immunoassays such as ELISA (enzyme-linked immunosorbent assay) identified unique biomarkers—cytokeratin-19 and tetranectin; cytokeratin-19 was upregulated in both the CIN 3 and CSCC IV conditions, and tetranectin was down-regulated in CSCC. Although CLU (clusterin) expressed upregulation, it couldn’t be statistically validated by ELISA [38]. Proteomic analysis of blood plasma has an ocean of biomarkers waiting to be discovered which can be further explored in developing better diagnostic candidates in cervical cancer. Future potential, on the other hand, lay in further optimizing plasma preservation protocols for blood-plasma-related body fluids, as well as creating simpler protein extraction techniques.

### 2.2. Serum Based Biomarkers

Among the numerous body fluids present in humans, serum includes a dynamic range of biomolecules that are far more critical in multiple intricate pathways implicated in various cancers, including cervical cancer (Table 2). We may change our approach to cervical cancer diagnosis and, potentially, therapeutic drug development by concentrating on a variety of serum fluid-based protein markers. TKT, FGA, APOA1, Survivin, TP53, CyclinB-1, and ANXA-1 were discovered as classifiers that play a critical role and were the proteomic serum biomarkers for cervical cancer that were found by collecting serum samples from cervical cancer patients prior to and following surgery, as well as from age-matched healthy control and cervical cancer patients [47]. FGA, a human fibrinogen which is synthesized in the liver, serves as a marker for a variety of tumor types [47]. FGA has also been linked to the pathophysiology of endometriosis (painful disorder emerging from an endometrial tissue wound) [47,57]. APOA1 is one such biomarker that has previously been discovered for endometrial and cervical high grade squamous intraepithelial lesions. It contributes to the transport of cholesterol from tissues to the liver where it is anabolized [47]. This protein was identified as a potential predictive serum marker for cervical cancer in this investigation [47]. Six potential serum-based proteins such as A1AT, PYCR2, TTR, ApoAI, VDBP, and MMRN1 are under the limelight as these are observed to be differentially expressed [49]. These proteins might be used as a set of biomarkers to distinguish cervical cancer patients from healthy controls and also between patient subgroups. Such studies give new hope for new ventures in this arena. In another study, conducted by label-based shotgun proteomics (iTRAQ) and targeted proteomics (MRM) techniques, significant changes in alpha-1-acid glycoprotein 1, alpha-1-antitrypsin, serotransferrin, haptoglobin, alpha-2-HS-glycoprotein, and vitamin D-binding protein were found [50]. Such evidence indicates that serum can be given higher emphasis as it is a reservoir of many proteins and markers, and can be considered as a warehouse of many cervical cancer biomarkers.

### 2.3. Mucous Based Biomarkers

Cervical crypts generate a viscous fluid from their secretory or gland cells, which is known as cervical mucus [58]. This fluid is a rich source of proteins belonging to two phases: aqueous and glycoprotein [28]. Cervical mucus is expected to contain proteins generated by both the lesion and the host in response to the lesion since it is formed in the milieu where cervical neoplasia develops. Because cervical mucus is essential for the health and maintenance of the female reproductive system, finding proteomic biomarkers in this fluid is a worthwhile aim. In the past, there have been few attempts to characterize the biochemical makeup of cervical mucus. The first research employed a SELDI-TOF MS analysis to assess criteria for mucous protein profiling [59]. Later, the human cervical mucus proteome was investigated using a variety of techniques, including one-dimensional and two-dimensional gel electrophoresis, liquid chromatography combined with mass spectrometry, such as the SELDI-TOF MS [28]. The comprehensive analysis detected a total of 107 unique proteins, among which a few have been previously linked to cervical carcinoma and pre-invasive diseases. They include annexin, tropomyosin, 14-3-3 sigma, calreticulin, and anterior gradient protein. Utilizing iTRAQ based labelled proteomics, another study from human cervical mucus found possible protein biomarkers that are differently expressed between cervical cancer patients and healthy controls [51]. The study found significant differences in 237, 256, and 242 proteins, respectively, amongst the comparable groups (endocervical adenocarcinoma vs. control, cervical adenocarcinoma in situ vs. control, and cervical adenocarcinoma in situ vs. endocervical adenocarcinoma) [51]. However, more research is needed to establish the fact that those differentially expressed proteins from mucus can actually be represented as biomarkers for the diagnosis and treatment of cervical cancer [51]. This research will pave the way for future discoveries of novel proteomics in CaCx biology which might be critical for developing improved diagnostics and targeted treatments, as well as immunotherapies and a variety of other approaches. Considering the huge potential advantages, more research efforts should be directed to developing cervical mucus-based proteomic biomarkers, given the little number of relevant and crucial biomarkers discovered thus far. Moreover, while conducting mucous-based biomarker development experiments, one has to very carefully choose the cohorts as the cervical mucus is hormonally responsive and composition of it will vary owing to menopause and the menstrual cycle.

### 2.4. Menstrual-Fluid Based Biomarkers

Menstrual fluid is commonly referred to as menstrual blood, however, it differs significantly from systemic blood in composition. It is a complex biological fluid made up of three different types of body fluids: whole blood, vaginal fluid, and uterine wall cells and their secretions [60]. While proteomic studies have revealed that menstrual blood and systemic blood share some protein indicators in common, there is evidence of few biomarkers exclusive to menstrual blood [61]. Multiple proteomic methodologies and analytical methods were used in a study by Yang et al., which resulted in the discovery of 385 proteins, unique to menstrual blood [60]. This work defined the proteomic composition of menstrual blood for the first time [60,61]. Additionally, this study concluded by emphasizing that menstrual fluid contains protein biomarkers valuable for a variety of illnesses, including cervical, breast, ovarian, and uterine cancers [60]. Despite the lack of menstrual blood-based proteomic research, one study discovered that polymorphisms in TAP (transporter associated with antigen processing) protein, which is important for the progression of high-risk HPV infections to cervical cancer, can be found in the menstrual blood of patients with high-risk HPV and cervical squamous intraepithelial lesion [52]. TAP 1 and 2 proteins play an important role in cervical cancer and targeting these proteins for the development of therapeutic medications would be one of the appropriate courses of action [62,63]. One disadvantage of relying on menstrual fluid is that it is not a readily available sample because not all females menstruate regularly. In particular, young females who have not attained puberty, women who have reached menopause, and women with other gynecological complications cannot rely on the menstrual blood-based diagnosis. Despite these shortcomings, these investigations have provided confidence and direction for developing agents for the protein biomarkers identified in menstrual fluid. Hence, conducting many more proteomic analyses of menstrual fluid and large-scale studies could make way for breakthrough discoveries in battling cervical cancer and many other cancers.

### 2.5. Cervicovaginal Fluid-Based Biomarkers

Cervicovaginal fluid (CVF) is one of the most important body fluids to investigate for indicators of cervical cancer. Secretions from sweat, sebaceous, Bartholin’s and Skene glands, plasma (transudate through the vaginal walls), exfoliated cells, bacterial byproducts, cervical mucus, fluids from the endometrium and the oviduct, and secretions from immune cells present in the vaginal wall make up cervicovaginal fluid [27,29,64]. The microenvironment of the cervix and the vagina, as well as the hormone cycle, influence the composition of cervicovaginal fluid over time [28]. CVF is a proximal fluid, which means it is more sensitive to the cervical and vaginal environment than other body fluids [29]. Because the CVF comes into direct contact with cancerous lesions in the cervix and vagina, it has the highest concentration of biomarkers related to cervical cancer [65]. Hence it has strong potential to aid in early detection and disease assessment.

In a very recent study by Gutierrez et al., candidate biomarkers for CIN2+ lesions were identified from dried CVF samples [66]. The researchers looked at the possibilities of utilizing mass spectrometry to identify protein biomarkers in dried self-sampled cervicovaginal fluid deposited on FTA cards. They discovered 18 proteins to be significant among a total of 207 proteins in their discovery cohort (CRNN, DDX3X/DDX3Y/DDX4, DESP, DHB4, DSG3, ELAF, GBP6, K1C14, K1C16, K2C1, LEG7, PKP1, PKP3, PLAK, SPR1A, SPR1B, SPR2A, and TGM1). Finally, the study suggested a seven-protein multivariate prediction model with sensitivity and specificity of 0.90 and 0.55, respectively [66]. Some of the most promising cervicovaginal fluid-based biomarkers for cervical cancer found until now are ACTN4, VTN, ANXA1, ANXA2, CAP1, and MUC5B. A study conducted by Starodubtseva and colleagues used a label-free quantification methodology based on LC-MS/MS method to perform proteome analysis. They discovered 27 proteins that were substantially expressed in cervical cancer patients across the four phases of samples [53], including the five proteins listed above. ACTN4, or alpha-actinin-4, is a promising biomarker candidate in cervicovaginal fluid for detecting cervical cancer in its precancerous stage. The presence of ACTN4 in CVF as a potential biomarker was initially confirmed in a study conducted by Raemdonck et al. in 2014, which clearly distinguished healthy controls and cervical cancer samples [54]. Higher carcinogenesis and development of cervical cancer are linked to increased ACTN4 expression. In vivo tumor growth and proliferation were also decreased when ACTN4 was knocked down [67]. Another group of biomarkers discovered in CVF was the pyruvate kinase M1/M2 isozyme [1]. These proteins belong to a family of glycolytic enzymes that play a crucial role in the cell’s energy supply. It has two isoforms, M1 and M2 [54,68]. PKM2 is overexpressed in cancer [60], hence by looking at its levels in CVF we can detect cervical cancer in its early stages. Vitronectin, or VTN, is another potential protein biomarker for cervical cancer in CVF. It belongs to the integrin family and to the family of glycoproteins. VTN is involved in cell–cell adhesion, cell motility, opsonization, and tumor metastasis [69]. ANXA1 is another protein biomarker that bears the potential to be used as a biomarker of cervicovaginal fluid [53]. ANXA1 (annexin 1) is a phospholipid-binding protein that is well known to inhibit the innate immune response and mediate efferocytosis to further regulate inflammation [70]. Due to the intimate interaction of CVF with malignant tissue, the concentration of ANXA1 rises in CVF, making it an accessible CVF based biomarker, much like the previous two biomarkers. All of the biomarkers described have been linked to cell–cell adhesion, cell proliferation, angiogenesis, metastasis, and many other processes. Although these proteins are prevalent in most malignancies, their presence in cervicovaginal fluid suggests the existence of cancer-promoting activities in the cervical and uterine regions. In both nonpregnant and pregnant women, cervicovaginal fluid (CVF) is a rich source of clinical information concerning the female reproductive system. None of the existing regular tests can determine the risk of neoplasia progression, which is a very important determination for women of reproductive age. We believe biomarkers identified from CVF can thereby open the route to decipher various stage-wise information at a diagnostic level.

### 2.6. Urine Based Biomarkers

Urinary proteomic biomarkers have an edge over the other body fluid-based biomarkers due to their non-invasiveness, availability, and high thermal stability [3,24]. In addition to the nitrogenous excretory metabolites, water, salts, and electrolytes in abundance, urine also consists of glomerular filtrate of plasma, hence the urinary proteome reflects the significant proteome changes at different and distant sites in the body [71]. Conventional proteomic methodologies for urinary proteome analysis involve 2D-DIGE-MS, LC-MS, SELDI-TOF, and CE (capillary electrophoresis)-MS, and validation of biomarkers by ELISA [72]. A study by Chokchaichamnankit et al. in 2019, compared urine samples from healthy people and cervical cancer patients at various stages. The study found 60 upregulated proteins and 73 downregulated proteins among the two groups, the majority of which were involved in blood coagulation and fibrinolysis [55]. When further validated by Western blotting, five proteins were found to be potent classifiers: leucine-rich-2-glycoprotein (LRG1), isoform 1 of multimeric protein (MMRN1), serpin B3 (SERPINB3), S100 calcium-binding protein A8 (S100A8), and a cluster of differentiation (CD-44). In separate research by Aobchey et al. in 2013, urine proteome analysis was carried out comparing healthy people and CaCx patients using ultrafiltration (3 kDa), standard 2D-DIGE, and LC-MS/MS to enrich low molecular weight proteins. The potential urinary biomarker associated with cervical cancer identified was the endorepellin LG3 fragment (25 kDa) [73]. The urine proteins discovered in this study were mostly involved in maintaining cell adhesion in the extracellular matrix (ECM), and their dysregulation aided carcinoma cell metastasis. Although we do see the potential for urinary biomarkers for identifying cervical cancer, there are still hurdles to overcome. These include the lack of a standard for sample processing and the difficulty in identifying low molecular weight proteins (less than 10 kDa). As urine includes significant levels of urea and other compounds such as toxins, excess water, and carbohydrates, it is crucial to develop a robust approach for sample processing and proteome extraction while removing those contaminants.

## 3. Body Fluid-Based Biomarker Development for Cervical Cancer: Scope and Difficulties

### 3.1. The Use of Unusual Body Fluids in the Search for Cervical Cancer Biomarkers

The changed proteome profiles exhibited in different body fluids might assist in biomarker identification since carcinogenesis affects the metabolic and signaling pathways to cater to the neoplastic growth of tumor cells. Blood plasma [13,38], serum [47,49], cervicovaginal fluid [29,53,54], cervical mucus [28], and urine [55,56] are the body fluid samples so far investigated for cervical cancer biomarker identification. The search for cervical cancer biomarkers in body fluids might be expanded to include uncharted territory such as amniotic fluid, saliva, cerebrospinal fluid, and most probably, bile. Saliva is one of the most easily and readily accessible body fluids, whose proteome is reflective of the blood plasma proteome [74]. Although possible differential biomarkers between control and cancer group samples have been discovered, such biomarkers are yet to be validated. In rare cases of cervical cancer in pregnancy (CCIP), amniotic fluid may potentially indicate differentially expressed proteins that might serve as biomarkers [75]. More study into body fluids might yield a powerful proteome biomarker cocktail that can help us in recognizing the disease one step ahead.

### 3.2. Current Clinical Challenges of Performing Omics-Based Studies including Cervical Cancer Patients

There are several challenges in implementing omics-based evaluation in the early detection and management of cervical cancer. In the theme of early detection, the main challenges involve implementing omics-based approaches at the population level wherein early detection has the most relevance. Early detection in cervical cancer has been extraordinarily successful with screening techniques such as pap smear [76], HPV detection by various methods, and visual inspection by acetic acid (VIA) [77,78]. These techniques have high sensitivity, reasonably high specificity, ease of implementation, are affordable, and have substantially reduced the incidence and mortality of invasive cervical cancer in many parts of the world, especially developed countries. However, pap smear and HPV detection have been difficult to implement at the population level in developing countries because of the requirement of expert cytologists and high costs, respectively. Moreover, collection, storage, and transport of samples has been challenging. When it comes to biomarker identification, omics-based techniques have outperformed the industry standard in terms of sensitivity, specificity, sample collection, storage, and transit convenience, all while lowering human expertise and costs. We hope that biomarkers discovered using such high-throughput technologies will also aid in the development of more sensitive and easy-to-use diagnostic tools in the future.

Some of the challenges enumerated above also apply to the use of omics-based techniques as prognostic and/or predictive biomarkers in the management of cervical cancer patients. However, there are additional considerations in this setting. The most important one relates to prospective banking of biomaterial in clinically well-annotated longitudinal patient cohorts, including those in clinical trials. Statistically, potentially useful biomarkers are included in multivariable interaction models (such as Cox models) to decipher whether any patient subgroups benefit or not from particular treatments. Omics-based biomarker discovery programs would have to establish the feasibility of long-term storage of body fluids, standardization on protocols, and prospective incorporation into clinical trial designs to be relevant to clinical decision making. With the advent of expensive biological treatments such as immune checkpoint inhibitors [79], drug-antibody conjugates [80], and others in the treatment of cervical cancer, there are substantial opportunities and a need for biomarker discovery in this disease.

## 4. Metadata Analysis from Proteomic Studies on Cervicovaginal Fluid (CVF)

A thorough literature review was undertaken on published or preprint studies up until 10 October 2020, where the search words were (“Cervical cancer proteome profiling”, “Cervicovaginal fluid”, “body fluid”, “Proteomics and cervicovaginal fluid”, “Proteomics of Cervical cancer body fluid”). The selected studies include the cervicovaginal fluid (CVF)-based proteomics studies. Due to CVF being local to the cervical microenvironment, we decided to conduct a meta-analysis of the data we could obtain from the papers we referred to while writing that section. We conducted a network analysis using both the STRING database and Cytoscape.

### 4.1. Network Analysis via STRING

STRING is a database created through a collaboration of the Swiss Institute of Bioinformatics, the Novo Nordisk Foundation CPR, the University of Zurich, and EMBL Heidelberg [81]. It is a web-based tool for analyzing and visualizing organism-wide protein networks [82].

#### 4.1.1. STRING Settings

On the STRING website, there are different input elements for various types of analyses. “Multiple Proteins” was the one employed in this study. The list of upregulated proteins was compiled from the texts and tables of three articles by Van Raemdonck et al., Starodubtseva et al., and Van Ostade et al. [53,54,65] (Figure 2A). The final list includes 47 proteins, some of which were repeated since they appeared in two or more articles. Using the Human Protein Atlas website, the proteins were examined for their existence in cervical cancer. This list was then saved as accession IDs on STRING. The following parameters were applied to perform the STRING analysis. We applied K-means clustering and the maximum number of clusters was set to 10. In the analysis section, the statistical background was set to “Whole Genome”, which is another standard feature of STRING. The results show all known and predicted associations between the input proteins, including both physical interactions as well as functional associations (Figure 2C).

#### 4.1.2. STRING Results

Apart from the image, the tables describing the biological processes (Appendix A), molecular functions (Appendix A), cellular components (Appendix A), KEGG pathways (Appendix A), reactome pathways (obtained from STRING analysis) (Appendix A), and the list of subcellular localizations (Appendix A) in the network were also obtained. Some of the biological processes (Appendix A) we found from the upregulated protein list from CVF were platelet degranulation, neutrophil degranulation, glycolytic processes, negative regulation of blood coagulation, regulation of cell death, regulation of anatomical size, protein polymerization, actin filament organization, and many more. We also were able to obtain four KEGG pathways—complement and coagulation cascade, viral carcinogenesis, glycolysis and gluconeogenesis, and vascular smooth muscle contraction— that were associated with the upregulated CVF proteins.

### 4.2. Network Analysis via Cytoscape

Cytoscape [83] along with the stringApp [84] plug-in was utilized for visualizing protein–protein interactions of the significantly upregulated proteins curated from an ensemble of papers studying cervicovaginal fluid proteomes. These networks were elucidated from data derived from three studies (as mentioned above) which fit a general theme of upregulation or exclusive presence in affected (cancerous, HSIL, LSIL, pre-cancerous) samples in comparison to the healthy controls. We believe network or mapping studies of such an ensemble will be a helpful source to draw biological inferences about the biologically enriched pathways and the involvement of their constituent proteins in the development and progression of cervical cancer.

#### 4.2.1. Cytoscape Settings

Cytoscape [83] and the stringApp [84] plug-in was utilized for visualizing protein–protein interactions in a radial layout (Figure 2B). More opaque lines represent a stronger level of confidence in the interaction between two proteins, represented by nodes. Sub-clusters within the network consisting of proteins showing high interconnectedness were extracted via the MCL algorithm (functionality provided by the stringApp plug-in) with an inflation parameter of 10.

#### 4.2.2. Cytoscape Results

Our meta-analysis on Cytoscape (Figure 2C) identified four prominent clusters from the network, each of which consist of four or more component proteins identified via cervicovaginal fluid-driven proteomics. These were as follows: a fibrinogen cluster identified by three chains -alpha, beta, and gamma (FGA, FGB, FGG), a complement cascade cluster (genes such as CD59, C9, SERPING1), a large cluster of actin-related proteins (composed of six ACT-genes) and a glycolysis cluster (genes such as PKM2, PGK1, and PGAM being key along with ATP5B- a player in oxidative phosphorylation).

### 4.3. Combined Interpretation of the Pathway Analysis Results

These protein groups interrelate with our STRING analysis results which yield multiple enriched pathways with strong links to cervical cancer, cancer in general, and its progression. For instance, the discovery of viral carcinogenesis in KEGG pathways (Appendix A) supports the hypothesis that HPV (human papillomavirus) infection plays a role in cervical cancer incidence. This adds more weight to the fact that research into HPV vaccines could be invaluable in bringing down the overall rate of cervical cancer across the world. However, recent cancer vaccine research has found that the objective response rate is typically low and can be attributed to a lack of powerful adjuvants for a sustained and durable immune response [85]. TLR (Toll-like receptor)-signaling pathways have been found to be targets of interference by HPV infection and subsequent cervical cancer progression [86]. We thereby suggest further research in this area to center around TLR-activating molecules and agonists (regulation of TLR by endogenous ligands found enriched in Appendix A), which can serve as adjuvants for the development of even more efficient cervical cancer vaccines.

The KEGG analysis from STRING had Complement and Coagulation Cascade as the top hit for enriched pathways. These can also be correlated from the enriched reactome pathways results from STRING (Appendix A) which includes Formation of Fibrin Clot (Clotting Cascade), Complement Cascade, as well as hemostasis. Abnormalities in the regulation of coagulation (as Cancer Coagulopathy) and fibrinolysis have long been linked to cancer as they directly influence tumor angiogenesis and thereby contribute to malignant growth [87]. We have also identified the three-component fibrinogen cluster in the Cytoscape Network (Figure 2C). Targeted pharmacological and/or genetic inhibition of pro-angiogenic activities of the hemostatic system are under study as potential anti-cancer treatments already. This needs further research to fine-tune these towards developing cervical cancer therapeutics in particular. On the diagnosis front, we note that ACTN4 has been found to contribute to the EMT (epithelial-to-mesenchymal transition) and tumorigenesis of cervical cancer [67]. The overexpression of EMT markers has been connected with poor prognosis and a high probability of metastasis [88], hence we outline the importance of ACTN4 and other actin-related proteins (part of the Cytoscape Cluster in Figure 2C) as potential biomarkers for monitoring progression.

## 5. Discussion

High-throughput proteomics assisted by bioinformatic methods of statistical analysis yields proteins differentially regulated in cancerous cells. LC-MS stands as one of the most robust techniques that can be employed to undertake differential proteomics (the discipline that detects proteins associated with a disease by means of their altered levels of expression between the control and disease states) [89]. Current proteomics methods, especially LC-MS involves complex sample preparation steps, advanced instrumentation, and costly consumables per sample. LC-MS also utilizes immense computing power for the quantification of proteins and their analysis [89]. These are some of the drawbacks and limitations of LC-MS that future optimization and innovation need to remedy. Proteomic methods are not only used to identify biomarkers but are also used to assess the interaction between the protein that is the therapeutic target, and the drug that help us to improve its affinity, efficiency, and efficacy [9]. The Shotgun method of biomarker identification is used during the discovery phase; however, biomarker validation should be done on a larger population utilizing focused approaches. Before a biomarker can be utilized clinically, it must go through a rigorous procedure that includes hypothesis formation, sample collection, data collecting, data analysis, assay development, assay validation, and regulatory approval. One of the most difficult aspects of biomarker discovery is the lack of well-established validation procedures for potential biomarkers identified from diverse body fluids.

Fluids like blood plasma and serum show large transient variabilities [3,90]. CVF changes due to changes in the menstrual cycle [91], sexual intercourse [29,92], urine due to altered diet [72], etc. Therefore, without a correct experimental technique, the inferences drawn from studies could lead to a lack of reproducibility and inappropriate findings. To prevent such complications, a large cohort must be used in a study to justify statistical significance. There is a necessity of expensive preprocessing such as the removal of high abundance proteins in blood or the enrichment of low abundance proteins in urine [3], aside from the social stigma associated with CVF sample collection. Along with the discovery of non-invasive, body fluid-based biomarkers for early detection of cervical cancer, spreading awareness and knowledge to clear up the still existing socio-cultural and psychological barriers associated with cervical cancer (HIV- related stigma) could encourage the early detection of cervical cancer by large numbers.

We have largely spoken about proteomic research done on diverse body fluids from CaCx patients in this review. However, the CVF studies were prioritized. CVF is a body fluid that fills the cervix’s niche, and changes in the fluid reflect the image of cervical cancer illness and development through its phases.

## 6. Conclusions

Employment of proteomics-based techniques in the analysis of body fluids relevant to cancer of the cervix holds the key to the “black box” of prognostic biomarkers, which are slowly being discovered by researchers all over the globe. The information extracted from the same is advantageous not only from the point of view of therapeutics but also from the perspective of drug design and discovery to target various differentially expressed protein biomarkers. Because the proteome of the cell is constantly changing as genes are turned on and off, it is an excellent approach for understanding various disease pathways and biochemical processes connected with certain disorders [89]. The tumor microenvironment has been found to have the highest abundance of cancer-related indicators in body fluids [5].

Tumor biomarkers are useful tools for doctors to use in early diagnosis, therapy response prediction, and disease monitoring. Blood-derived biomarkers for cervical cancer suffer from several of the disadvantages listed above. Other body fluids, such as CVF and, in particular, urine, have previously shown good results and may prove to be more specific to the cervix, making them appealing media for biomarker identification and clinical application of validated markers. Amniotic fluid, for example, has a comparable potential that is currently untapped. The discovery and acceptance of new biomarkers as clinically useful is predicted to have a significant influence on cancer research, impacting the detection and treatment of different cancers, including cervical cancer. A thorough understanding of each biomarker will be necessary for accurately diagnosing the condition and guiding the selection of the best therapy options.

## Figures and Tables

**Figure 1 proteomes-10-00013-f001:**
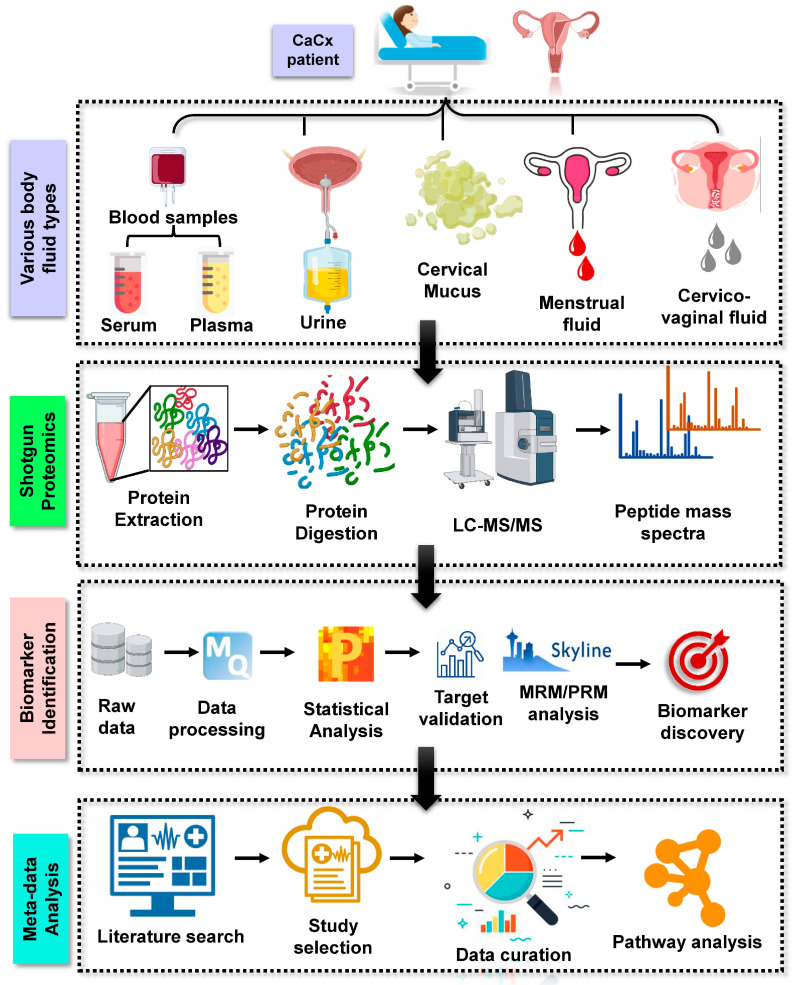
Depicting the general method for proteomic analysis of cervical cancer derived from various body fluids. Blood (plasma and serum), urine, cervical mucus, cervico-vaginal fluid, and menstrual fluid are the several types of body fluids from which proteomics investigations have been undertaken to date. Shotgun proteomics involves protein extraction from these body fluids followed by digestion, LC separation, and mass spectrometry analysis. In addition, raw MS data is statistically analysed, followed by target validation (MRM/PRM) and biomarker discovery. Such analysed datasets from the literature could also be used for computational-based metadata analysis to discover cumulative illness development pathways.

**Figure 2 proteomes-10-00013-f002:**
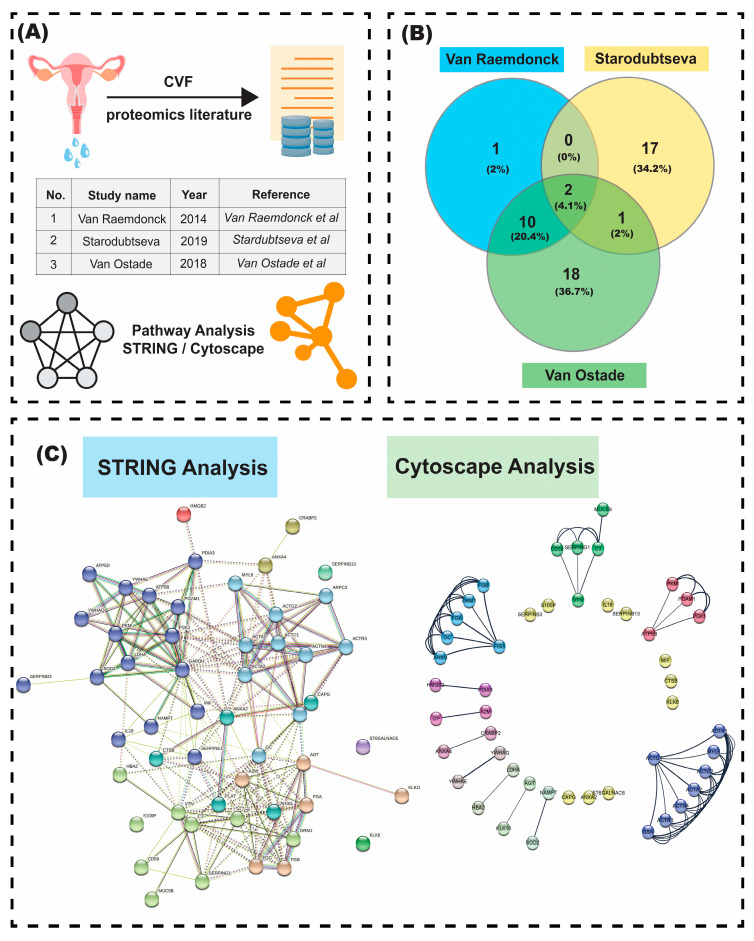
(**A**) Workflow used in the metadata analysis derived from several proteomic studies including cervicovaginal fluid; (**B**) Venn diagram showing the number of unique and common protein biomarkers mentioned in three different pieces of literature; (**C**) The image obtained after doing K-means clustering in STRING shows six distinct and well-clustered networks and four outliers that could not be clustered. It also includes interactors that were found from the analysis conducted on the STRING webserver. The network has 52 nodes, 215 edges, an average node degree of 8.27, and a PPI enrichment *p*-value of <1.0 × 10^−16^. The Protein-Protein Interaction network visualized using Cytoscape and analyzed via stringApp gave four prominent clusters (highly interconnected clusters consisting of four nodes or more). The radial layout was used to depict the sub-clusters in a clean and concise format along with the coupled proteins (seven clusters of two nodes) and singletons (10) as outputted by the MCL-clustering algorithm (inflation parameter = 10).

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
