# Peer review of "Insights on Proteomics-Driven Body Fluid-Based Biomarkers of Cervical Cancer"

_proteomes, 2022, doi:10.3390/proteomes10020013_

Round 1
Reviewer 1 Report
The article describes the many different biomarkers for cervical cancer and/or its precancerous stages that have been identified in several body fluids including serum and plasma, menstrual blood, urine, mucus and cervicovaginal fluid. To my knowledge these data have not been reviewed before and in that respect the article could be of interest for researchers and practitioners in the field. Overall, the first descriptive part is comprehensive and informative, however the second part (metadata analysis) clearly needs more work.
The article is about body fluids, however it is clear that the emphasis is on cervicovaginal fluid, especially towards the end. This should also be stated in the title. Some parts are also repeated several times, e.g. general considerations about the use of body fluids, and this should be avoided. I would also suggest to distinguish more between diagnostic and therapeutic applications, often the two are used through each other in the same paragraph.
For chapter 2, every paragraph is finished with a remark about the need for more proteomics work on these body fluids, but there is no specification or suggestion for further work given in these concluding remarks, hence they are of limited value.
I hereby list my comments according to the position in the text:
Line 62: the authors state that the omics approach is capable of profiling a huge number of samples and there is no need for labeling. I do not completely agree with that statement. (Prote)omics technologies can process much less samples compared to e.g. ELISA (but they give an overwhelming amount of information), and many proteomics techniques use labeling for quantification.
Line 67: opens the path…
Line 69: in cancer patients…
Line 79: the FIGO stages are not explained
Line 82: this is confusing since it is about intracellular contents, not about biofluids
Line 90: please make sure all abbreviations are explained in the listing on p1 or are at least once explained in the text
Line 100: approaches, which is one…
Line 132: there are plenty of studies about cancer biomarkers in body fluids so why were these chosen? Some of the papers are not very recent and could maybe be replaced by more recent ones.
Line 134: present in that microenvironment…
Line 137 and 145: which body fluid?
Line 152: repetition elsewhere
Line 208-210: I am missing the study of Berggrund et al. (doi: 10.1074/mcp.RA118.001208)
Line 231-232: indicate…
Line 264: from endometrial “would”? Must be “wound”?
Line 271: overexpressed in which sample types?
Line 283-286: why is it ideal then? I would rather think it is less ideal because of larger variation.
Line 288-293: this sentence is not very clear to me, please re-write.
Line307-310: redundancy, this kind of sentences is repeated too much
Line 330-331: references?
Line 356-365: As can be seen on the date of publications it was the group of Van Raemdonck et al. who first proposed the ACTN4 biomarker in CVF and later on this was confirmed by the group of Starodubtseva, not the opposite.
Line 368-369: please give reference of the CVF paper wherein this was described
Line 374: integrin family and to the family of glycoproteins.
Line 376-377: please give reference of the CVF paper wherein this was described
Line 342-389: I am missing the publication of Gutierrez et al. who performed proteomics on dried CVF samples (DOI: 10.3390/cancers13112592)
Line 389-390: water, salts, and electrolytes
Line 407-412: difficult to read, please re-write
Line 417: ways to cater the neoplastic
Line 422: The search for cervical cancer biomarkers
Line 426: reference
Line 445-451: not clear what the authors want to say. Do they suggest omics techniques for diagnostics? That will be much more expensive than e.g. PCR!
Line 458: patient subgroups benefits (or not) from…
Line 458-459: “These biomarkers are overwhelmingly performed”… not clear what is meant with that
Line 431-468: it is not clear what the authors want to say in this paragraph. The paragraph describes tissues, not biofluids and the difference between diagnosis and therapeutics is not clear.
Line 469: this paragraph is not necessary since it overlaps with other parts of the article. I suggest to leave it out.
Line 508-515: there is more emphasis on CVF in the searches. This is nowhere mentioned or explained. The results are only based on the results of three articles from two groups, which limits the value of this work.
Line 528-529: which protein network image, please refer to figure or list.
529-530: unclear, please re-write
Line 530-538: these supplementary tables are not very informative. Many of these programs can convert the tables into graphs which would be much clearer.
Line 558-559: which are these proteins? There is no correlation with table S1
Line 560: I can only see three clusters
Line 561: “It also includes the interactors that were found from the STRING analysis”. Seems evident because it is a string analysis
Line 539-567: There are many unanswered questions that arise from this chapter. What is the conclusion from this study? Which pathways can be identified? How can this help in diagnosis/biomarker identification?
Line 568-644: parts of this chapter belong to the previous paragraphs (meta-analysis) and the rest is very redundant to the other paragraphs.
Supplementary table 1:
Criteria of selection:
- Van Raemdonck et al.: where is the p-value referring to?
- Van Ostade et al.: where are the two tables that are mentioned?
Author Response
NOTE: The revisions to the manuscript have been made in the track-change mode and are therefore highlighted. We request you to follow the track-change version in order to recognize the precise line-by-line revisions as mentioned in our point-by-point responses below.
The cover letter has the entire response written in it.

Reviewer 2 Report
Manuscript proposed by Mukherjee and co-workers (proteomes-1653387) entitled “Insights on Proteomics-driven body fluid-based biomarkers of Cervical cancer” is a review article presenting details of the body fluids-based biomarkers that have been experimentally found as particular to cervical cancer. In my opinion, this is an interesting and well written article.
My comments are presented below.
Major concerns:
- Abstract – lines 28-30 - Here we present our perspective on proteomics for biomarker discovery in this paper by addressing several pertinent problems, including the challenges that we confront - please define clearly the which group of biomarkers were described in the paper.
Due to the fact that mass spectrometry and liquid chromatography coupled with mass spectrometry are the leading techniques in such kind of study, the paragraph presented the advantages and also limitations of these methods should be included.
Additionally, the paragraph summarizing the analytical techniques used in the analysis of presented biomarkers should be presented.
The discussion about the sensitivity of the used analytical techniques should be presented in relation to the possibility of using these methods in medical diagnostics.
Abbreviations should be explained. For example, MRM, PRM iTRAQ and others are presented in the text without explaining them.
Check the reference style.
Make changes in the text.
Check and correct English
Author Response

(The authors gave the same response as above.)

Round 2
Reviewer 1 Report
Most of the manuscript has improved markedly, hence the authors have already done a nice job. I also agree with most of the author’s responses; however I think there are still some aspects that could be clearly improved and these relate to my comments 45 and 46 of my previous review. This concerns the explanation of the different features and pathways found by STRING/KEGG and Cytoscape in the discussion, which, to my opinion, is too late.
For sections 4.1 and 4.2 the authors write in their rebuttal letter: “Section 4.2, however, is simply a section describing the methodology used with Cytoscape. Interpretation of those results has been discussed in the discussion section”.
I do not agree with this view because the discussion section in a review paper is usually very general and is not about new results.
Therefore I would strongly recommend to put most of the results from the metadata analysis not in the discussion, but in the metadata analysis section and to better structure that part, for instance:
- STRING analysis
1.1. Settings
1.2. Results
- Biological processes
- Molecular functions
etc...
- Cytoscape analysis
2.1. Settings
2.2. Results
- Common features (overlapping results between both methods)
In addition I have some minor comments:
Line 460: high-throughput
Line 475: Metadata Analysis from proteomic studies
Supplemental tables: graphs must be corrected : when the pathways are indicated on the X-axis, they do not need to be mentioned in the legend. Also some colors are not explained in the legend.
Author Response
Uploaded as a WORD file
